# Evolutionary changes in transcription factor coding sequence quantitatively alter sensory organ development and function

Simon Weinberger[1,2,3], Matthew P Topping[4,5], Jiekun Yan[1,2], Annelies Claeys[1,2], Natalie De Geest[1,2], Duru Ozbay[1,2], Talah Hassan[4], Xiaoli He[4], Joerg T Albert[4,5,6], Bassem A Hassan[1,2,3,7]*, Ariane Ramaekers[1,2,7]*

[1]VIB Center for the Biology of Disease, VIB, Leuven, Belgium; [2]Center for Human Genetics, University of Leuven School of Medicine, Leuven, Belgium; [3]Program in Molecular and Developmental Genetics, Doctoral School for Biomedical Sciences, University of Leuven School Group Biomedicine, Leuven, Belgium; [4]Ear Institute, University College London, London, United Kingdom; [5]Centre for Mathematics and Physics in the Life Sciences and Experimental Biology (CoMPLEX), University College London, London, United Kingdom; [6]Department of Cell and Developmental Biology, University College London, London, United Kingdom; [7]Institut du Cerveau et de la Moelle Epinière (ICM) - Hôpital Pitié-Salpêtrière, UPMC, Sorbonne Universités, Inserm, CNRS, Paris, France

*For correspondence: bassem. hassan@icm-institute.org (BAH); ariane.ramaekers@icm-institute. org (AR)

Competing interests: The authors declare that no competing interests exist.

**Abstract** Animals are characterized by a set of highly conserved developmental regulators. Changes in the *cis*-regulatory elements of these regulators are thought to constitute the major driver of morphological evolution. However, the role of coding sequence evolution remains unresolved. To address this question, we used the Atonal family of proneural transcription factors as a model. *Drosophila atonal* coding sequence was endogenously replaced with that of *atonal* homologues (*ATHs*) at key phylogenetic positions, non-*ATH* proneural genes, and the closest homologue to ancestral proneural genes. *ATHs* and the ancestral-like coding sequences rescued sensory organ fate in *atonal* mutants, in contrast to non-*ATHs*. Surprisingly, different ATH factors displayed different levels of proneural activity as reflected by the number and functionality of sense organs. This proneural potency gradient correlated directly with ATH protein stability, including in response to Notch signaling, independently of mRNA levels or codon usage. This establishes a distinct and ancient function for ATHs and demonstrates that coding sequence evolution can underlie quantitative variation in sensory development and function.

## Introduction

Animals share a toolkit of highly conserved genes governing key processes of development and homeostasis. Differential deployment of these genes, caused by *cis*-regulatory sequence variation, is often considered to be the key driver of developmental evolution (*Carroll, 2008*; *Rokas, 2008*; *Prud'homme et al., 2007*; *Wittkopp and Kalay, 2012*; *Wray, 2007*). However, coding sequence (CDS) changes could also play an important role and the relative, or differential, contribution of *cis*-regulatory versus CDS variation to developmental evolution is under debate (*Cheatle Jarvela and Hinman, 2015*; *Hoekstra et al., 2007*; *Lynch and Wagner, 2008*; *Stern and Orgogozo, 2008*).

Transcription factors (TFs) of the basic helix-loop-helix (bHLH) superfamily regulate key aspects of cell differentiation throughout metazoans (*Ledent et al., 2002*; *Skinner et al., 2010*). Proneural bHLH TFs belong to three distinct families named for their founding members: Atonal, Neurogenin and Achaete-Scute. They play a central role in neurogenesis by conferring neuronal identity onto ectodermal cells. This process involves the Notch signaling pathway – another 'toolkit component' - which restricts expression of proneural TFs and acquisition of neural fate to neuronal precursor cells. This refinement is thought to be governed mostly at the transcriptional level (*Barad et al., 2011*; *Bertrand et al., 2002*; *Guruharsha et al., 2012*; *Hartenstein and Stollewerk, 2015*; *Quan and Hassan, 2005*; *Vervoort and Ledent, 2001*). However, in recent years, accumulating evidence pointed toan important role for post-transcriptional mechanisms in regulation of proneural protein activity (*Guillemot and Hassan, 2017*). In particular, several reports suggest that Notch signaling can regulate proneural function via modulation of proneural protein stability (*Kiparaki et al., 2015*; *Qu et al., 2013*; *Sriuranpong et al., 2002*).

In the fruit fly, the proneural TF Atonal (Ato) confers neuronal identity onto a subset of photoreceptors of the compound eye, the R8-cells, and onto a subset of mechanosensory receptors, the chordotonal organs (ChOs) (*Jarman et al., 1995*, *1994*, *1993*). Ato homologues (ATHs) have been identified throughout bilateria (*Simionato et al., 2007*). Interestingly, sense organs mediating vision, hearing, gravity and proprioception are specified by Ato and its homologues in flies, mice and other animals, suggesting a common origin of these organs (*Arendt et al., 2002*; *Hassan and Bellen, 2000*; *Quan and Hassan, 2005*). Here, we use ATHs as a model to investigate the potential contributions of TF CDS evolutionary variation to sensory organ development and function. We selected ATHs from key bilaterian groups and analyzed their capacity to substitute for Ato function in the fruit fly *Drosophila melanogaster*. To specifically study the effects of CDS variation, the *Drosophila ato* open reading frame was substituted by its homologues in the endogenous locus, leaving *cis*-regulatory sequences intact. Our results show that ATHs share functional properties that distinguish them from proneural factors of the Neurogenin and Achaete-Scute families. These properties likely arose in the common ancestor of the Ato/Neurogenin superfamily. We find that coding sequence differences between ATHs are associated with quantitative changes in proneural activity, that is, in the number of sensory precursor cells that they specify. By measuring mechanotransduction in the Johnston's organ (JO), a cluster of ChOs in the second antennal segment, as a quantitative test of sensory organ performance, we demonstrate that different ATHs rescue various aspects of JO functionality. Finally, we find that changes in proneural activity are likely mediated by differential proneural protein stability in response to Notch–mediated lateral inhibition.

## Results

### ATHs represent a functionally distinct proneural group

ATHs share specific amino acids in the bHLH domain that distinguish them from other bHLH families (*Figure 1A*) (*Hassan and Bellen, 2000*; *Quan et al., 2004*). Using these amino acids as criteria, we identified ATHs throughout the bilaterian lineage but not in the genomes of other sequenced metazoans (*Hydra magnipapillata, Nematostella vectensis, Acropora digitifera, Amphimedon queenslandica, Sycon ciliatum, Trichoplax adhaerens, Mnemiopsis leidyi, Pleurobrachia bachei*), confirming a previous report on the bilaterian origin of the Ato family (*Simionato et al., 2007*). The published *Ato-like/ATH* genes in cnidarians, planarians and Amphioxus, did not appear in our analysis as proper *ATHs* (*Beaster-Jones et al., 2008*; *Martín-Durán et al., 2012*; *Richards and Rentzsch, 2015*; *Seipel et al., 2004*). However, we cannot rule out the possibility that these genes have diverged to a level that makes them unrecognizable as *ATHs* by sequence alone. Interestingly, 110 out of 119 *ATH* sequences corresponded to single exon genes indicating a strong constraint against the acquisition of introns and alternative splicing in *ATHs* (data not shown). ATHs from different phyla lack any conservation outside the bHLH protein domain. In addition, the position of the bHLH domain differs between ATHs. It can be found either in the middle or at the C-terminal part, but not at the N-terminal part, of the protein (*Figure 1C* and data not shown). We also observed several cases of independent duplications and losses of members of this family, confirming that the *ato* family belongs to the bilaterian developmental gene 'toolkit' and that its members have undergone extensive diversification.

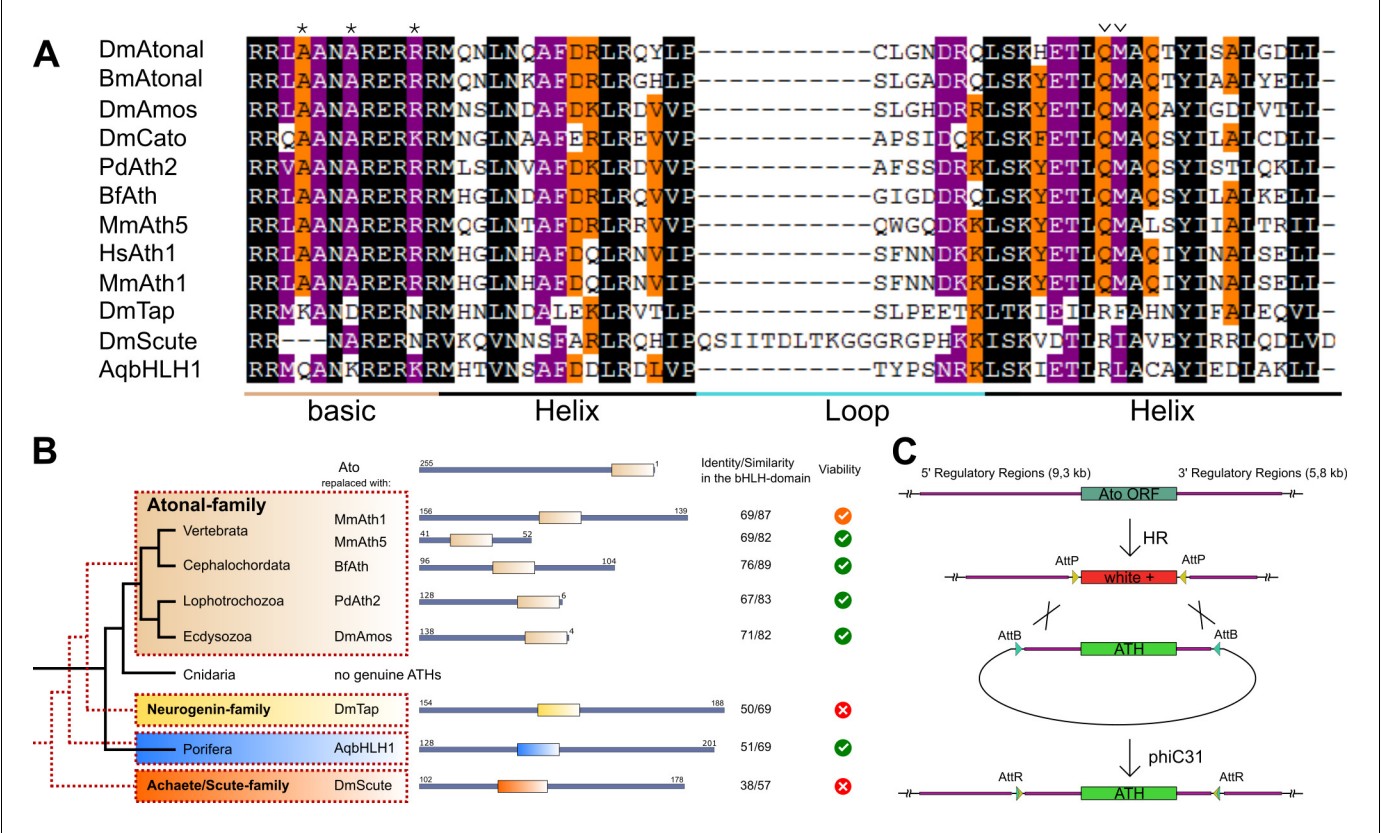

**Figure 1.** ATHs represent a functionally distinct proneural group. (**A**) Alignment of bHLH domains of the ATHs used in this study. The bHLH domains of the three proneural TF families are characterized by the presence of family-specific AA (marked with * and ˅). Asterisks: AA distinguishing ATHs from members of the Neurogenin family (**Quan et al., 2004**). (**B**) Phylogenetic position of the selected homologues and evolutionary relationship of the control proteins. The black tree shows the phylogeny of the organisms. The red dashed tree shows the relationships between protein families. The diagrams show the primary structures of the proteins; the bold box is the bHLH domain, numbers indicate the length of the protein outside the bHLH domain. Green tick: homozygous viable stocks , orange tick: semi-viable, red cross: absence of unbalanced flies. Dm: *Drosophila melanogaster*; Mm: *Mus musculus*; Bf: *Branchiostoma floridae*; Pd: *Platynereis dumerilii*; Aq: *Amphimedon queenslandica*. (**C**) Schema of the improved IMAGO approach (**Choi et al., 2009**). See also **Figure 1—figure supplement 1**.

The following figure supplement is available for figure 1:

**Figure supplement 1.** Upon replacement in *ato* endogenous locus, all *ATHs* and *AqbHLH1* rescue the lethality caused by loss of *ato* in the fruit fly.

For detailed analysis, we selected a subset of ATHs at key phylogenetic positions: the two mouse ATHs (MmAth1 and MmAth5), the lancelet BfAth (*Branchiostoma floridae*), the annelid PdAth2 (*Platynereis dumerilii*) and the fruit fly Ato paralog Amos, as well as representatives of the Neurogenin and Achaete-Scute proneural factor families as outgroup controls. We also included the ancestral-like AqbHLH1, a sponge TF equally related to the Ato and Neurogenin families (*Figure 1B*) (*Richards et al., 2008*).

The IMAGO (Integrase-Mediated Approach for Gene knock-Out) approach (*Choi et al., 2009*) was used to replace the *ato* open reading frame with that of the selected genes in the endogenous *ato* locus (*Figure 1C*). By placing the expression of the transgenes under the control of the endogenous *ato* regulatory sequences, the resulting Knock-In (KI) lines allow investigation into the effects of CDS variation independently of *cis*-regulatory changes.

All *ATH-KI*s and the ancestral-like *AqbHLH1–KI* were viable. In contrast, the KIs of *Drosophila Neurogenin* and *Achaete-Scute* family members were not (*Figure 1B*). The absence of *ato* endogenous sequence and lack of protein expression was confirmed for all viable KIs (*Figure 1—figure supplement 1A and B*). These results indicate that ATHs share functional properties that distinguish

them from other proneural bHLH TFs. Moreover, the rescue by the ancestral-like TF AqbHLH1 suggests that ATH–specific functional properties arose in the ancestral proneural bHLH TF(s). Interestingly, although MmAth1 was previously thought to be a fully functional Ato homologue (*Wang et al., 2002*), *MmAth1-K*I flies were only semi-viable.

## ATHs show quantitative variation in retinal precursor specification

Differentiation of the fruit fly retina starts at the beginning of the third larval instar stage (L3) as a wave that sweeps from posterior to anterior across the eye primordium in an indentation called the morphogenetic furrow (MF), leaving behind rows of regularly spaced R8 photoreceptor precursors (*Figure 2A*). Once specified, each R8 triggers the formation of an entire ommatidium, a unit of the compound eye (*Roignant and Treisman, 2009*). In the absence of Ato, R8s fail to form and, consequently, retinal differentiation is stopped resulting in the absence of ommatidia (*Jarman et al., 1994*).

To test whether *ATH* transgenes can substitute *ato* function during eye development, we analyzed retina differentiation in viable KI-lines. We labeled R8s with anti-Senseless (*Nolo et al., 2000*; *Pepple et al., 2008*) and all photoreceptors with anti-Elav (*Koushika et al., 1996*). In all cases, the gross morphology of the primordium was preserved and contained Senseless positive cells (*Figure 2B* and *Figure 2—figure supplement 1A*). However, differences in R8 spacing were detected between KI-lines. The R8 pattern generated by the expression of *ato* and *amos*, a fly *ato* paralog, were indistinguishable from wild-type (*Figure 2Ci*, and *Figure 2—figure supplement 1Bi-iii*). The *MmAth1-KI* eye primordium presented a dramatic increase in density of Senseless expressing cells (*Figure 2Cii*). The remaining KI-lines (*BfAth-KI*, *MmAth5-KI*, *PdAth2-KI*, *AqbHLH1-KI*) had some irregularly spaced R8s in the MF, with nuclei in close proximity (*Figure 2Ciii+iv* and *Figure 2—figure supplement 1Biv-vii*; white arrows). However, R8 spacing defects were progressively compensated for at later stages of retina differentiation with more posterior R8 cells presenting more regular spacing.

We noted that the size of the differentiated portion of the eye primordia appeared to be different between the KI-lines (*Figure 2B* and *Figure 2—figure supplement 1A*). To verify this, we counted the number of ommatidial rows along the anterior-posterior axis at the end of the L3 stage in the viable KI-lines. *ato-* and *amos-KI* retinae displayed the same number of ommatidial rows as the wild-type (~23 rows on average). In KI-lines with irregularly spaced R8s (*BfAth-K*I, *MmAth5-K*I, *PdAth2-KI*, *AqbHLH1-*KI), we counted significantly fewer rows (*Figure 2D*). The severe R8 patterning defects in eye-antennal discs of the *MmAth1KI* precluded counting ommatidial rows. However, at the end of the L3 stage the size of the differentiated portion of the *MmAth1-KI* eye primordium, characterized by Senseless expression, was not significantly different from that of the control lines (*Figure 2E* and *Figure 2—figure supplement 1C*). Importantly, the duration of the L3 stage, during which retinal differentiation takes place, did not differ between the lines (data not shown). These observations suggest differences in R8 specification rates between the KI-lines.

Once specified, the R8 precursor cell orchestrates assembly of an ommatidium (*Roignant and Treisman, 2009*). All viable KI-lines, except for *MmAth1-KI*, showed a correct temporal sequence of recruitment and full set of ommatidial cells (*Figure 2—figure supplement 2A–H*). In the *MmAth1-KI*-line, the eye primordium was severely mispatterned and markers of distinct photoreceptors, that are usually mutually exclusive, were often co-expressed (*Figure 2—figure supplement 2E*).

Finally, we found that changes in development translated into differences in morphology. The eyes of *ato-KI* and *amos-KI* were anatomically indistinguishable from wild-type (*Figure 2Fi* and *Figure 2—figure supplement 1Di-iii*), whereas the other KI-lines displayed a variety of eye phenotypes. The KI-lines with irregular arrangement of R8s in the MF (*BfAth-KI*, *MmAth5-KI*, *PdAth2-KI*, *AqbHLH1-KI*) had rough eyes, a few irregularly sized ommatidia and occasional additional ectopic interommatidial bristles (*Figure 2Fiii-iv* and *Figure 2—figure supplement 1Div, vi-viii*). Furthermore, KI-lines with the slowest rate of differentiation (*PdAth2-KI* and *MmAth5-KI*) displayed the smallest eyes. The increased R8 density in the MF was associated with an increase in ommatidia along the dorso-ventral axis resulting in higher total ommatidia numbers (*Figure 2G–I*). In line with their strong developmental defects, *MmAth1-KI* adults had severely mispatterned eyes precluding quantification of ommatidial content (*Figure 2Fii*). Most importantly, all retinae harbored cells expressing rhodopsin 6, an R8-specific photopigment (*Figure 2J* and *Figure 2—figure supplement*

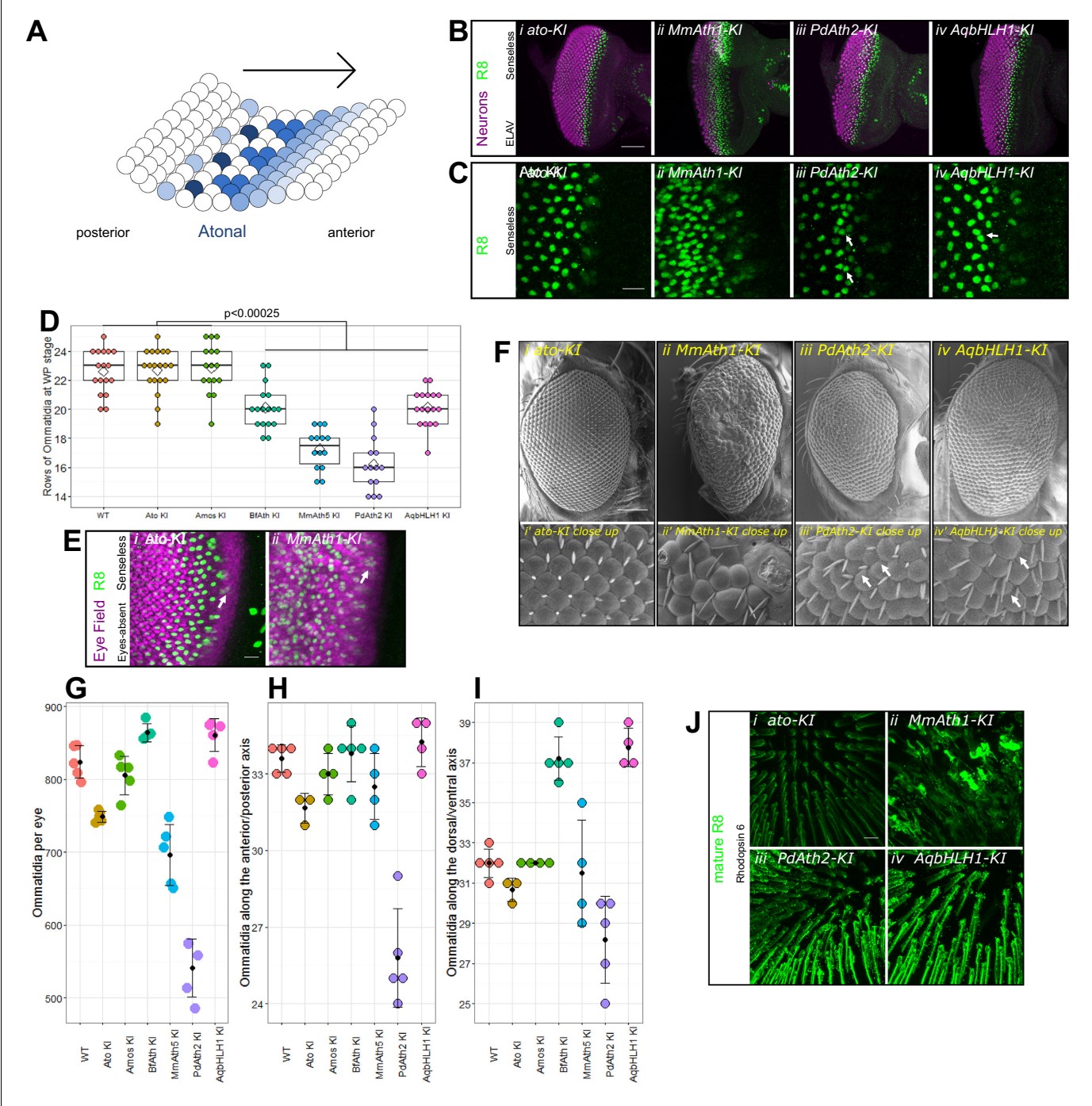

**Figure 2.** ATHs specify retinal precursors at different rates. (**A**) Ato specifies the R8 photoreceptors in a sequential manner (from posterior to anterior) in an indentation in the imaginal disc called the morphogenetic furrow (MF), leaving behind regularly spaced R8 precursors. (**B**) Anti-ELAV and anti-Senseless immunostainings on late L3 eye-antennal discs: (i) ato-KI, (ii) MmAth1-KI, (iii) PdAth2-KI, (iv) AqbHLH1-KI (scale bar 50 μm). (**C**) Anti-Senseless immunostainings at the level of the MF: (i) ato-KI, (ii) MmAth1-KI, (iii) PdAth2-KI, (iv) AqbHLH1-KI (scale bar 10 μm). (**D**) Number of rows of ommatidia consisting of at least six photoreceptors at white pupae stage; boxes indicate interquartile ranges, lines medians, diamonds means and whiskers data ranges. Summary statistics can be found in *Supplementary file 6A* and all p-values (t-test) for all pairwise comparisons in *Supplementary file 6B*. (**E**) Anti-Eyes-absent and anti-Senseless immunostainings on P0 eye-antennal discs: (i) ato-KI, (ii) *MmAth1-KI* (scale bar 50 μm). (**F**) Scanning electron microscopy images of compound eyes (i–iv) and close ups (i'–iv') of (i+i') ato-KI, (ii+ii') *MmAth1-KI*, (iii+iii') *PdAth2-KI*, (iv+iv') *AqbHLH1-KI* (scale bar 50 μm); arrows indicate smaller or larger ommatidia. (**G**) Ommatidia number of female compound eyes; black dots indicate means, error bars SEM. (**H+I**) Ommatidia numbers along the anterior/posterior axis (**H**) and (**I**) dorsal ventral axis of adult female compound eyes; black dots indicate means, error

*Figure 2 continued on next page*

*Figure 2 continued*

bars SEM. (**J**) Anti-hodopsin 6 immunostainings of adult KI fly retinas (i) *ato-KI*, (ii) *MmAth1-KI*, (iii) *PdAth2-KI*, (iv) *AqbHLH1-KI* (scale bar 50 μm); All confocal images (B, C, E, J) are maximum intensity projections. See also *Figure 2—figure supplements 1* and *2*.

The following figure supplements are available for figure 2:

**Figure supplement 1.** ATHs specify retinal precursors at different rates.

**Figure supplement 2.** Correct ommatidia differentiation takes place in all KI-lines except *MmAth1-KI*.

*1E*) (*Papatsenko et al., 1997*), suggesting that terminal differentiation of the R8 cells occurred properly.

These data show that coding sequences of the *ATHs* and of the ancestral-like *AqbHLH1* are able to specify R8 cells but display quantitative differences in R8 selection.

## ATHs show quantitative variation in chordotonal organ (ChO) specification

Ato is required for the specification of ChO precursor cells. Once formed, ChO precursors give rise to four cells via a stereotyped division pattern (*Brewster and Bodmer, 1996*; *Jarman et al., 1993*; *Lai and Orgogozo, 2004*) (*Figure 3A*). We focused on a cluster consisting of five ChOs in the larvae, a number reduced to one in the absence of Ato (*Jarman et al., 1995*, *1993*).

We found that all viable KI-lines formed larval ChO neurons, albeit with differences in their numbers (*Figure 3B*). As in the eye, both *ato-* and *amos-KI*-lines were indistinguishable from the wild-type. All the other KI-lines, except *MmAth1-KI*, displayed a reduced number of neurons, which varied from one to four. The different KI-lines behaved as an allelic series - *MmAth1* and *ato* and *amos* > *BfAth* > *AqbHLH1* > *MmAth5* and *PdAth2* - characterized by the formation of a decreasing number of ChOs. It is noteworthy that this is a similar order to that observed for the rate of R8 specification (R8: *MmAth1* and *ato* and *amos* > *BfAth* and *AqbHLH1* > *MmAth5* > *PdAth2*).

Next, we tested whether the formed ChOs contained the proper set of accessory cells (cap cell, scolopale cell and ligament cell) (*Figure 3A*). Immunostainings for cell-type specific markers revealed that this was the case for all viable KI-lines, indicating that, once specified, ChO precursors underwent the proper division pattern (*Figure 3C–D*, *Figure 3—figure supplement 1A–H*). In addition, the correct localization of Spacemaker, a component localized to the lumen between the neuron and scolopale cell, suggested differentiation of ChO neurons (*Husain et al., 2006*), *Figure 3C–Diii*, *Figure 3—figure supplement 1A–Hiii*).

These results show that ATH coding sequences and the ancestral-like AqbHLH1 share functional properties but display quantitative differences, specifically in ChO precursor selection.

## ChOs of Johnston's organ (JO) specified by different ATHs show quantitative functional variation

In the fruit fly, the Johnston's organ (JO), a cluster of ChOs in the second antennal segment, senses auditory cues, gravity and wind by detecting relative motions between the second and the more distal third antennal segments. ChO neurons that respond to different cues are characterized by distinct response properties (*Albert and Göpfert, 2015*; *Kamikouchi et al., 2009*; *Yorozu et al., 2009*). To analyze the function of the ChOs specified by Ato homologues, we carried out a detailed analysis of JO functionality in the different KI-lines.

Sound-evoked antennal deflections are actively amplified by neurons in a frequency and intensity-dependent way (*Göpfert and Robert, 2003*; *Göpfert et al., 2005*; *Nadrowski and Göpfert, 2009*). In Drosophilids, these active processes tune the fly's antennal sound receivers to species-specific frequencies while passive receiver properties are largely conserved (*Riabinina et al., 2011*). ato mutants lack ChOs of the JO and the rotational joint of the antennal sound receiver fails to form, causing the third antennal segment to be stiffly coupled to the second (*Göpfert et al., 2002*). All ATHs and AqbHLH1 restored the mobility of the antennal joint. With the sole exception of *PdAth2-KI* flies which were characterized by a stiffer antennal joint, the *ATH-KI* passive receivers were

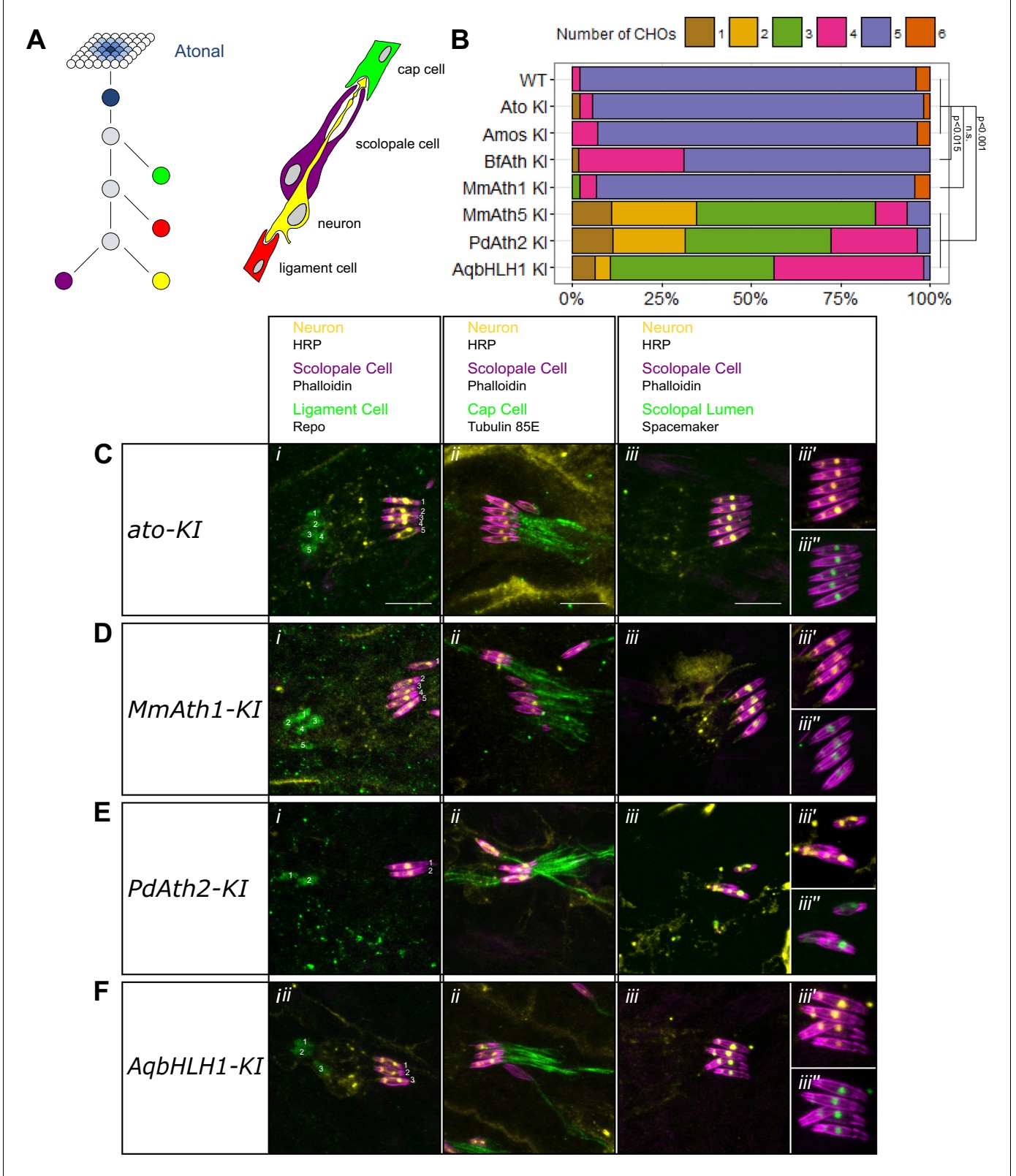

**Figure 3.** ATHs show quantitative variation in chordotonal organ specification. (**A**) Ato specifies the ChO precursor cell that gives rise to the four ChO cells via a fixed division pattern. (**B**) Frequency plot of the number of ChOs present in the embryonic lateral ChO cluster (14–17 hr after egg laying). Summary statistics can be found in *Supplementary file 6C* and all p-values (Fisher exact test) for all pairwise comparisons in *Supplementary file 6D*.

*Figure 3 continued on next page*

Figure 3 continued

(C–F) Cell content of embryonic lateral ChOs formed by the different ATHs. Neurons (yellow) are labeled using anti-HRP; scolopale cells (magenta) with Phalloidin labeling. In addition, ligament cells are revealed using anti-repo (i), cap cells using anti-tubulin 85E (ii) and the scolopale lumen using anti-spacemaker (iii), and immunoreactivity. (iii' + iii''): close ups of phalloidin labeling together with HRP (iii') or spacemaker (iii'') immunostainings. All images are maximum intensity projections. (C) *ato-KI*, (D) *MmAth1-KI*, (E) *PdAth2-KI*, (F) *AqbHLH1-KI* (scale bar 10 µm). See also *Figure 3—figure supplement 1*.
The following figure supplement is available for figure 3:

**Figure supplement 1.** ATHs and AqbHLH1 - derived chordotonal organs present correct cell content.

statistically indistinguishable from the controls (*Figure 4A*, top). In contrast, there was substantial variation in the active receivers (*Figure 4A*, bottom). Compared with the *ato-KI* controls, the active properties of both *amos-KI* and *BfAth-KI* displayed statistically identical best frequencies, whereas the receivers of *MmAth5-KI*, *MmAth1-KI* and *AqbHLH1-KI* were shifted to higher frequency values and - particularly prominent in *MmAth1-KI* - also displayed sharper frequency tuning. The active receivers of *PdAth2-KI* antenna were statistically identical to the passive ones, indicating that active amplification was not taking place. Calculating the energy gain by ChOs confirmed these findings (*Figure 4—figure supplement 1A*).

Next, we analyzed the mechanical properties of the ChOs by measuring their gating compliances (*Albert et al., 2007*). In wild-type ChOs, two distinct gating compliances are associated with two different types of mechanotransducers: (1) a sharp compliance around the resting position, linked to the gating of sensitive 'auditory' transducers, and (2) a shallower compliance, reflecting the gating of less sensitive, 'wind/gravity' transducers (*Effertz et al., 2012*). In all viable KI-lines, with the sole exception of the *PdAth2-KI*, both types of compliance are fully or partly detected, indicating that both sensory modalities are restored. The gating compliances of the *ato-KI*, *amos-KI* and *BfAth-KI* were indistinguishable from each other and virtually identical to the previously reported wild-type condition (*Effertz et al., 2012*). The gating compliances of the *MmAth5-KI*, *AqbHLH1-KI* and *MmAth1-KI* rescues displayed more substantial differences with respect to each other and to the control conditions, mostly residing in the components associated with wind and gravity perception. In contrast, properties of *PdAth2-KI* receivers suggested a specific loss of one type of transducer (*Figure 4—figure supplement 1B*).

Finally, we measured the nerve responses over a wider range of antennal displacements as a read-out of the electrical properties of ChO neurons (*Figure 4B*). Whereas the compound action potential responses of *ato-KI*, *amos-KI* and *MmAth1-KI* rescue flies were virtually indistinguishable, the responses of *BfAth-KI*, *MmAth5-KI*, *AqbHLH1-KI* and *PdAth2-KI* flies were substantially reduced in amplitude, with the responses of the *PdAth2-KI* flies being the smallest. Detailed analyses indicate that their differences may result from a changed force sensitivity (*Figure 4—figure supplement 1C*).

Put together, our data show that in all viable KI-lines, JO - the antennal ChO - is at least partly functional. ChOs of viable KI-lines display both mechanical (gating compliances) and electrical (nerve activity) signatures of mechanotransducer gating, and all but one had a fully mobile antennal joint. However, different ATHs rescue distinct aspects of JO function, which have previously been linked to distinct cellular subgroups and distinct mechanosensory submodalities.

## Differential ATH proneural potency, mRNA levels and autoregulation

Like other proneural TFs, Ato regulates its own transcription (*Baker et al., 1996*; *Sun et al., 1998*). Therefore, the phenotypic variation observed between the KI-lines could simply derive from the different capacity of ATHs to act on the fruit fly *ato* regulatory sequences, leading to different levels of their own expression. If true, we would predict that *MmAth1* would show higher mRNA expression levels compared with *amos*, which in turn would show higher mRNA expression than *MmAth5*. To test this hypothesis, we compared the mRNA levels of each *ATH* in developing eye-antennal discs (*Figure 5—figure supplement 1A&B*). We found that all *ATHs* displayed higher expression levels than *ato* with no correlation between mRNA levels and phenotype, indicating that differential autoregulation alone cannot explain quantitative phenotypic variation between them. To confirm these observations and quantify the functional differences between ATHs and AqbHLH1 independently of

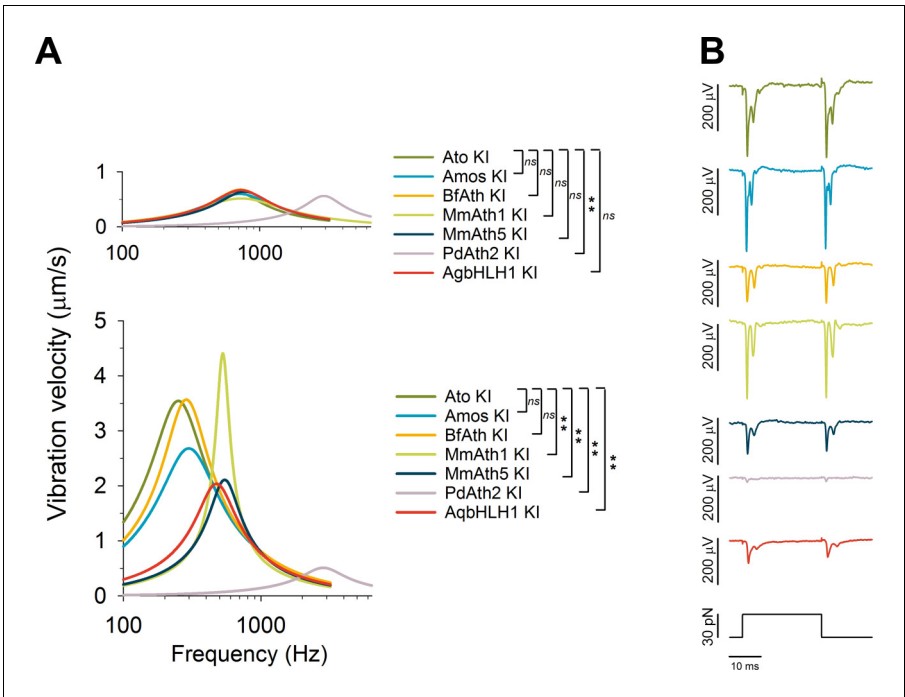

**Figure 4.** ChOs of Johnston's organ (JO) specified by ATHs and AqbHLH1 are functional and different functional aspects are restored in a quasi-modular way. (**A**) All viable KI-lines, at least partly, restored the rotational joint of the flies' antennal sound receivers (top: metabolically inactive – passive - receivers of $CO_2$-sedated, $O_2$-deprived flies; bottom: metabolically active receivers of awake, $O_2$-supplied flies). The free mechanical fluctuations of the unstimulated receivers reveal the specific antennal best frequencies for all ATHs (color legend to the right indicates statistically significant differences compared with the *ato-KI*). Whereas the passive receivers (top) are virtually identical in all *ATH-KI* rescues (with the exception of *PdAth2-KI*), the active receivers (bottom) display characteristic differences in frequency tuning and auditory amplification. Summary statistics and p-values (t-test or Mann-Whitney Rank Sum) can be found in ***Supplementary file 6Q and R***. (**B**) In response to a medium-range force step (30 pN corresponding to ~ half-maximal activation of *ato-KI* controls), all antennal nerves of viable KI-lines produce a symmetric compound action potential (CAP) response to both the onset and the offset of the antennal deflection, as reported for wild-type controls. The amplitudes of the CAP responses, however, vary among the different KI-lines, with *amos-KI* and *MmAth1-KI* rescues reaching *ato-KI* control levels and the remaining ATHs producing significantly smaller CAPs (*BfAth1 > MmAth5 > AqBHLH1 > PdAth2*). See also ***Figure 4—figure supplement 1***.

The following figure supplement is available for figure 4:

**Figure supplement 1.** Analysis of dynamic stiffness of the antenna and the nerve responses over a wide range of stimuli revealed a quasi-modular JO function in the KI-lines.

autoregulation, we used the Gal4/UAS system, in which the expressed ATHs would be unable to transcriptionally regulate their own transgenes. All *UAS-ATH* and *UAS-AqbHLH1* transgenes were inserted at the same genomic locus, and overexpressed along the anterior-posterior boundary of the wing imaginal disc, in a classic proneural activity assay. The numbers of ectopic sensory organs formed on overexpression of the *ATHs* were then quantified (***Figure 5A***). These ranged from 0 to 3 ectopic bristles in the case of *UAS-PdAth2* to around 65 for *UAS-MmAth1* (***Figure 5B***). Ato and Amos overexpression resulted in distinct numbers of bristles (p<0.001; Wilcoxon signed rank test) while multiple *UAS-ato* experiments were statistically indistinguishable. Interestingly, no ectopic bristles were induced by *MmAth5* ectopic expression. Importantly, the number of ectopic sense organs specified by each ATH correlated with their effect during ChO and R8 development (***Table 1***). Thus,

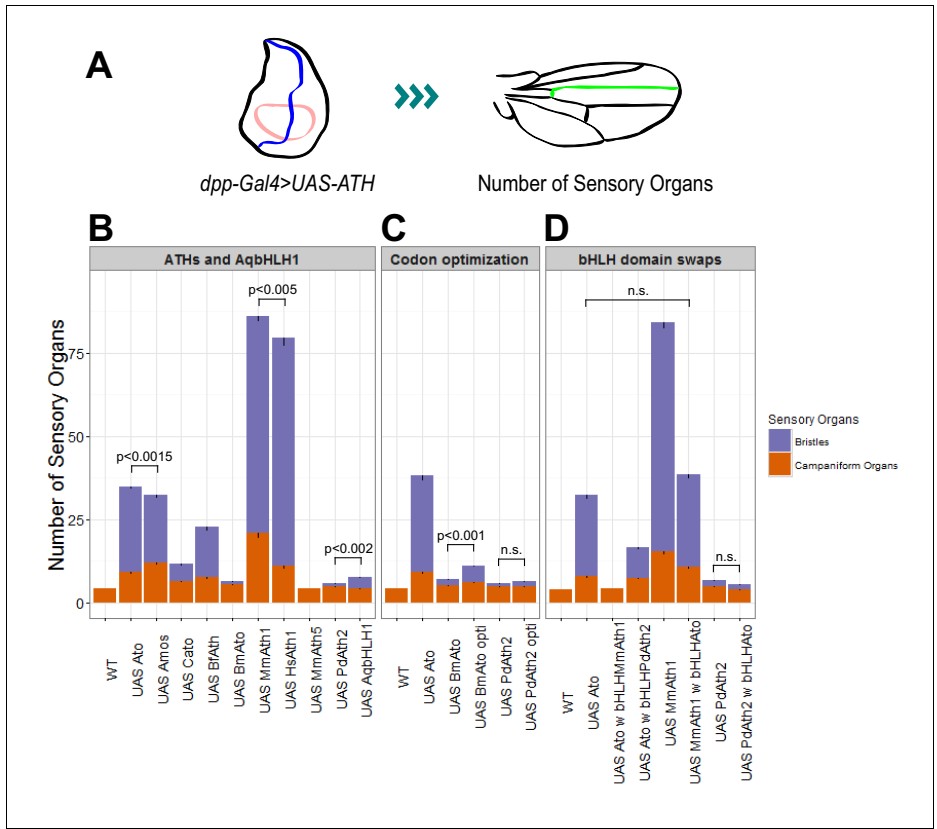

**Figure 5.** *ATHs* ectopic expression induces formation of distinct numbers of ectopic sensory organs. (**A**) Schematics of the wing disc assay; ectopic sensory organs formed upon ectopic expression of *ATHs* and *AqbHLH1* driven by *dpp-Gal4* were counted along the mid-vein until and including the anterior cross-vein. (**B–D**) Numbers of ectopic sensory organs; data are presented as mean with SEM. Statistics: Wilcoxon rank sum test on the number of bristles except for *MmAth1* and *HsAth1* on number of campaniform organs. Summary statistics can be found in **Supplementary file 6E,F,I,J,M and N** and all p-values (Wilcoxon rank sum test) for all pairwise comparisons in **Supplementary file 6G,H,K,L,O and P**. See also **Figure 5—figure supplement 1**.
The following figure supplement is available for figure 5:

**Figure supplement 1.** Expression of *ATHs* and *AqbHLH1* is upregulated compared with *ato*.

quantitative variation in ATH capacity to specify sensory organ precursors must be an intrinsic property of their CDS – which we henceforth term proneural potency.

Next, we used the ectopic expression assay to perform a series of control experiments. Ectopic expression of the *ato* paralog *cato* led to formation of a reduced number of bristles compared with *ato* and a*mos*, ruling out that the *Drosophila* origin of *amos* and *ato* by itself explains their stronger performance compared with most *ATHs* (**Figure 5B**). MmAth1's strong proneural potency in this

**Table 1.** Comparison of proneural potency across tissue.

| Tissue | Order |
| --- | --- |
| R8 | MmAth1 ≈ Ato ≈ Amos > BfAth ≈ AqbHLH1 > MmAth5 > PdAth2 |
| ChO | MmAth1 ≈ Ato ≈ Amos > BfAth >> AqbHLH1 > PdAth2 ≈ MmAth5 |
| Ect. Ex. | MmAth1 >> Ato > Amos > BfAth >> AqbHLH1 > PdAth2 > MmAth5 |

system was confirmed by the similar results obtained by overexpressing its close homolog, the human *HsAth1* (*Figure 5B*). We also tested ectopic expression of the silk moth *BmAto*. *Bm*Ato mRNA was shown to be expressed in the moth developing eye disc in a pattern reminiscent of the fruit fly *ato*, suggesting that BmAtois a true Ato functional homologue (*Yu et al., 2012*). Interestingly, as shown previously by Yu and colleagues, *BmAto* ectopic expression induced formation of few bristles, demonstrating that proneural potency varies even between functional homologues (*Figure 5B*). Importantly, changes in codon usage biases across *ATH* coding sequences do not underlie variation in proneural potency as ectopic expression of highly codon optimized versions of the two ATH genes characterized by the least favorable codon adaptation index, *BmAto* and *PdAth2* (*Supplementary file 4*), resulted in a marginal (*BmAto*) or non-significant (*PdAth2*) increase in the number of ectopic bristles (*Figure 5C*).

Finally, we tested whether proneural potency was determined by the bHLH domain only or, alternatively, was a distributive property of the entire protein. We swapped bHLH domains between Ato and ATHs with strong (MmAth1) or weak (PdAth2) proneural potency. We found that exchanging the bHLH domain always resulted in a decrease or complete loss of proneural potency (*Figure 5D*). Thus, proneural potency does not appear to be an intrinsic property of the bHLH domain but likely emerges as a property of the entire ATH protein.

## ATH protein steady state dynamics correlate with proneural potency and are regulated by Notch signaling

*ATH* ectopicexpression in the wing disc revealed that variation in proneural potency between ATHs cannot be explained solely by differences in transcriptional regulation. We thus investigated whether ATH factors were differentially regulated at the post-transcriptional level. To compare ATH protein levels independently of changes in mRNA expression, we overexpressed selected *ATHs* for which antibodies were available, that is *ato*, *amos*, *cato* and *MmAth1* together with a destabilized GFP transgene (*UAS-dGFP*) (*Lieber et al., 2011*) during wing disc development. Anti-ATHs immunostainings revealed clear differences in protein expression between the four ATHs (*Figure 6A*). Specifically, MmAth1 protein was detected in most dGFP expressing cells, Ato and Amos presented intermediate expression levels, and Cato protein was detected in only a few dGFP positive nuclei. ATH proteins being detected in different subsets of dGFP positive nuclei indicates that they are subject to differential post-transcriptional regulation. Importantly, differences in protein expression between the four ATHs correlated with their proneural potency (*Figure 4B*), suggesting that this is influenced by the ATH protein steady state dynamics. Restriction of proneural gene expression from cell clusters to single, or a few, neural precursor cells is a conserved process during sense organ development. It depends on Notch-mediated lateral inhibition and is thought to be largely regulated at the transcriptional level (*Barad et al., 2011*; *Guruharsha et al., 2012*), whereby Notch signaling induces the transcriptional silencing of proneural genes. However, a few reports indicate that regulation of proneural activity by Notch signaling could also involve post-transcriptional mechanisms, including regulation of protein stability (*Kiparaki et al., 2015*; *Qu et al., 2013*; *Sriuranpong et al., 2002*). We hypothesized that differential sensitivity to Notch-dependent regulation of protein stability might explain differences in proneural potency among ATHs. To test this idea, we overexpressed an active form of Notch ($N^{ICD}$) in a pattern orthogonal to ATH overexpression in the wing disc. In this assay, activated Notch should not transcriptionally silence *ATH* expression from the heterologous UAS promoter. Remarkably, we observed that while $N^{ICD}$ efficiently reduced Amos protein levels, it had little or no effect on MmAth1 (white arrows, *Figure 6B*). Similar results as with Amos were also obtained with Ato and Scute (*Figure 6—figure supplement 1A*). These indicate that variation in proneural potency among ATHs is at least partly mediated by differential ATH protein steady state levels in response to Notch signaling. Next, we tested whether protein stability differences between ATHs could be detected during normal development. We compared mRNA and protein localization during retinal differentiation of *ato-KI* and *MmAth1-KI* eye-antennal discs. As expected, mRNA and protein showed a striped expression pattern at the level of the MF (*Figure 6—figure supplement 1B and C*). Importantly, a*to* and *MmAth1* mRNA stripes were of similar width and showed the same steep decline at the posterior limit of the MF. In contrast, MmAth1 protein persisted more posteriorly (*Figure 6Bi+i' and iii+iii'*) and, unlike Ato, did not resolve into single cells, revealing a higher stability of MmAth1 compared with Ato (*Figure 6B ii+ii' and B iv'+iv'*). We sought to test other ATHs under similar endogenous conditions by generating KI-lines with

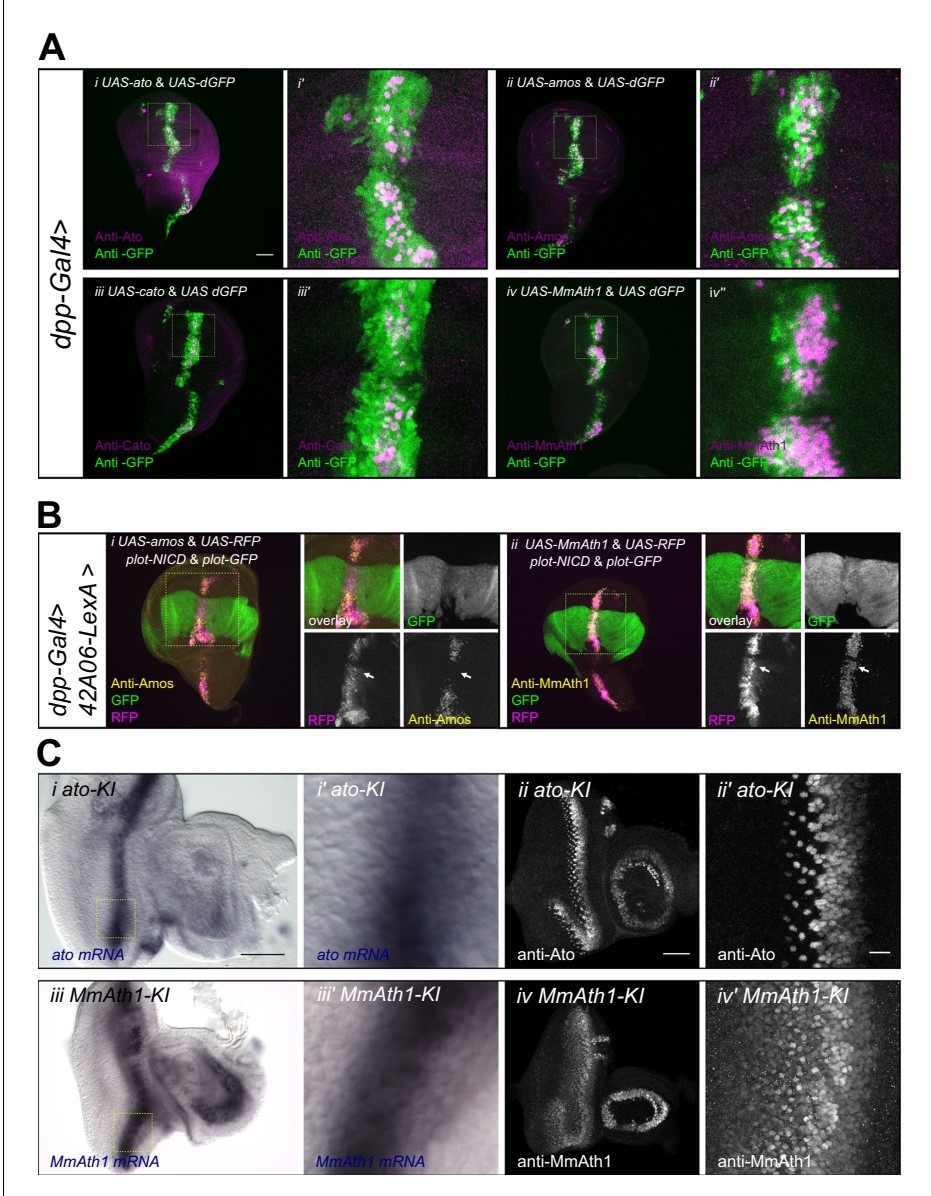

**Figure 6.** ATH protein steady state dynamics correlate with proneural potency and are regulated by Notch signaling. (**A**) Wing disc ectopic expression and immunostainings of (i+i') *ato* and *dGFP*, (ii+ii') *amos* and *dGFP*, (iii +iii') *cato* and *dGFP*, (iv+iv') *MmAth1* and *dGFP*. (i+ii) overviews (scale bar 50 µm), (i'+ii') close ups. (**B**) (i) Anti-Amos; (ii) anti-MmAth1 immunostainings and endogenous fluorescence of RFP and GFP in L3 wing discs. *ATHs* and *RFP* are ectopically expressed along the dorso-ventral axis (*dpp*-GAL4) while *42A06*-LexA drives the expression of *GFP* and $N^{ICD}$. (**C**) Comparison of the expression levels of *ato* and *MmAth1* mRNAs and proteins; (i+i') *ato in situ* hybridization in an *ato-KI* eye-antennal disc; (i) overview (scale bar 50 µm) and (i') close up indicated by square in (i). (ii+ii') Anti-Ato immunostaining in an *ato-KI* eye-antennal disc; (ii) overview (scale bar 50 µm); (iii) close up at the MF (scale bar 10 µm). (iii+iii') *MmAth1 in situ* hybridization in a *MmAth1-KI* eye-antennal disc; (iii) overview and (iii') close up indicated by square in (ii). (iv+iv') Anti-MmAth1 immunostaining in a *MmAth1-KI* eye-antennal disc.; (ii) overview; (iii) close up at the MF. Confocal images in A, B and C (ii+ii'; iv, iv') are maximum intensity projections. See also *Figure 6—figure supplement 1*.

The following figure supplements are available for figure 6:

**Figure supplement 1.** ATH mRNA and protein expression.

**Figure supplement 2.** C-terminal tags differently affect ATH function.

C-terminal GFP-tagged versions of Ato, MmAth1, MmAth5 and PdAth2. To our surprise, the addition of the GFP tag alone altered the phenotypes KI-lines compared with the untagged versions. In addition, the effect of the tag was variable depending on the ATH. Ato::GFP behaved like the non-tagged protein both in terms of expression during retinal development and morphology of the adult eye (*Figure 6—figure supplement 2Ai-i''*). The same was true for Ato::mCherry (see below). In contrast, multiple injections of a *MmAth1::GFP* transgene resulted in pupal lethality in $F_0$, suggesting strong dominant effects. *MmAth5::GFP-KI*-lines displayed dominant phenotypes in the adult eye similar to that of *MmAth1-KI* (*Figure 6—figure supplement 2Aii+ii'*), and were homozygous lethal. *PdAth2::GFP-KI*-lines displayed a rough eye phenotype (*Figure 6—figure supplement 2Aiii+iii'*). Next, we sought to examine the protein dynamics of the tagged-ATHs during retina development. To this end, we crossed the *ato::mCherry-KI* line with the GFP-tagged KI lines of *ato*, *MmAth5* and *PdAth2*, respectively (*Figure 6—figure supplement 2Ai''+ii''+iii''*). Ato::GFP and Ato::mCherry proteins had identical expressiondynamics (*Figure 6—figure supplement 2A'''*). Consistent with the *MmAth1*-like dominant phenotype of *MmAth5::GFP-KI* line, MmAth5::GFP was more stable than Ato::mCherry, that is extended more posteriorly (*Figure 6—figure supplement 2Aii'''*). Surprisingly, PdAth2::GFP protein expression was not detected in the heterozygous eye-antennal discs(*Figure 6—figure supplement 2Aiii'''*), but was weakly detectable in homozygosity (*Figure 6—figure supplement 2B*). The variability of the effect of the tagon ATH function, as shown by the range of eye phenotypes displayed by the corresponding KI-lines, makes it difficult to draw conclusions about the expression patterns of the ATH::GFP proteins. However it is interesting to note that the dominant *MmAth1-KI*-like phenotype of the *MmAth5::GFP-KI* flies correlates with longer perdurance of MmAth5::GFP protein compared to Ato::GFP, further supporting the notion that proneural potency is linked to steady state proneural protein levels.

## Discussion

Using the Ato family of proneural TFs as a model, we explored the contribution of transcription factor coding sequence variation to the evolution of sense organs within the bilaterian lineage. Combining *in vivo* CDS exchanges with locus-controlled ectopic expression, this study presents, to our knowledge, the first direct functional comparison of closely related developmental genes across key phylogenetic positions.

When expressed in the fruit fly, the ATHs and the sponge AqbHLH1, but neither Achaete-Scute nor Neurogenin-related TFs, successfully specified Ato-dependent sensory organs. Thus, this cell fate determination decision is executed by Ato family members only, establishing the Ato family as a distinct proneural family with unique functional properties. Conservation of these properties in the sponge ancestral-like ATH AqbHLH1 suggests that they were already present in ancestral ATHs. Interestingly, unlike other ATHs, AqbHLH1 is also able to perform Neurogenin–like functions (*Quan et al., 2004*; *Richards et al., 2008*). This supports the notion that subfunctionalization eventstook place during the early expansion of the bHLH superfamily (*Richards et al., 2008*; *Simionato et al., 2007*).

We demonstrate that variation in *ATH* CDS is associated with changes in sensory organ number. In this study, we referred to the capacity of each ATH to specify different numbers of sense organ precursor cells as their proneural potency. Domain swap experiments suggest that proneural potency is an emergent property of the entire protein, rather than a property of a specific domain or motif. This is consistent with results from domain swap in experiments between DmAto and its functional homolog BmAto (*Yu et al., 2012*). Interestingly, for a given ATH, proneural potency was largely consistent across sensory organ sub-type, both endogenous and ectopically induced, indicating that it constitutes an inherent property of each ATH. We observed two exceptions to this observation indicating some degrees of tissue-specificity. First, ectopic expression of multiple independently generated *UAS-MmAth5* lines did not result in ectopic bristles. Second, MmAth1 strong proneural potency was manifest during retinal development and upon ectopic expression but not during formation of the larval ChOs.

The *Drosophila* JO – the fly's 'inner ear' – consists of functionally and molecularly specialized neuronal subpopulations, which mediate the sensations of distinct mechanosensory submodalities, such as for example wind, gravity or sound (*Albert and Göpfert, 2015*). As the *Drosophila* JO does not receive efferent feedback from the brain (*Kamikouchi et al., 2010*), a detailed analysis of its function

can directly reveal the stimulus transduction and amplification properties of its chordotonal neurons. Our experiments show that although all ATHs are able to restore functional JOs and ChO nerve activity, they affect various functional parameters differently and independently. Some ATHs, like MmAth1, only affect the properties of the 'insensitive', that is, wind or gravity sensing, transducers, whereas others, like MmAth5, only affect the *molecular properties* - but not the predicted *numbers* - of transducer channels. Moreover, the specific phenotypes that result from different ATHs present a degree of complexity (see supplemental *Supplementary file 6Q,R and S* for details), which makes it rather unlikely that a simple change in neuronal numbers is their sole cause. If the effects were only caused by an increase (or decrease) in neuronal numbers, then a substantial overall reduction in transducer channels, as seen for example in *AqbHLH1-KI* flies, would reflect a proportional reduction in neurons. As JO neurons contribute ~75% of the antenna's total passive stiffness (*Göpfert et al., 2005*), the antennae of *AqbHLH1-KI* flies should be considerably softer, but - in contrast - their steady state stifnesses are significantly higher (by almost a factor of two). The antennae of *MmAth5-KI* flies have unchanged numbers of transducer channels but substantially increased stiffness values. In conclusion, the coding sequence motifs of the here tested *ATHs* appear to both generate and specify JO subpopulations in a submodality-specific manner, suggesting that specification of mechanosensory submodality is at least partly dependent on the *ato* coding sequence. We speculate that *ATH* coding sequences could interact differentially – either quantitatively or qualitatively or both – with the Ato downstream regulon. As a result, the distinct mechanosensory submodalities would emerge from parallel, and partly independent, 'channels' within the Ato downstream regulatory network.

Our results suggest that differences in proneural potency between ATHs can be explained, at least in part, by differential protein steady state dynamics. Moreover, our data are consistent with the notion that differential ATH protein stability is expressed as differential responses to Notch activity. Together with previous evidence for proneural protein degradation in response to Notch signaling (*Kiparaki et al., 2015*; *Qu et al., 2013*; *Sriuranpong et al., 2002*), this suggests a model by which functional differences between ATHs both in development and evolution could be caused by changes in sensitivity to Notch-mediated protein degradation. In our study, MmAth1 is characterized by a particularly strong proneural potency, leading to formation of supernumerary sense organ precursors compared with Ato itself, both during retinal development or as a consequence of ectopic expression in the wing. Unlike Ato protein, the expression of MmAth1 in the MF or upon overexpression in the wing shows little or no restriction to discrete cells. Interestingly, this is reminiscent of MmAth1 expression during mouse cerebellum development (*Wang et al., 2005*), suggesting that the reduced sensitivity to Notch-mediated protein degradation could be an inherent property of MmAth1. MmAth1 stability is also regulated by BMP (*Zhao et al., 2008*) and Shh-signaling (*Forget et al., 2014*). Put together, this suggests that regulatory evolution could extend beyond *cis*-encoded transcriptional variation to include post-translational changes, such as variation in the regulation of protein stability by distinct signaling pathways, encoded at the level of the coding sequence.

Finally, most ATHs behaved as hypomorphic alleles in our assays. As it is rather unlikely that a variety of organisms have suboptimal proneural ATHs, we postulate a model in which the CDS, and within the CDS, the conserved bHLH and non-conserved domains-, *cis*-regulatory elements, protein interaction partners and their post-translational regulation co-evolve to ensure robustness of ATH-dependent processes. It seems reasonable to assume that such mechanisms are not unique to ATHs or proneural proteins, but rather extend to other TF families and signaling molecules.

## Material and methods

### Bioinformatic survey across published genomes

The bHLH domain of Ato was used as a query for tBLASTn search (protein query against the translated genome) in genomes of organisms at key node positions in the metazoan lineage (listed in *Supplementary file 1*). The significant hits were inspected for the presence of ATH-specific AA (*Quan et al., 2004*). The longest open reading frame containing the bHLH domain was predicted from the sequence data and compared with the annotation when available.

## Acquisition of DNA

DNA sources are listed in *Supplementary file 2*.

## Generation of the new IMAGO-allele

A new *ato^w* IMAGO allele was generated using the method described by *Choi et al. (2009)* consisting of ends-out homologous recombination with the pWhiteStar vector. AttP-sites were inserted ~1.5 kb upstream and ~0.5 kb downstream of the *ato* ORF to prevent possible interference with the core promoter region. The missing up- and downstream regions were re-supplied during the recombinase mediated cassette exchange.

## Transgenic fly stocks

Transgenes were generated using standard molecular biological techniques. bHLH TF coding sequences were cloned into pUASTattB (*Bischof et al., 2007*) and into the IMAGO vector (*Choi et al., 2009*). N^ICD was cloned into a pLOTattB vector (*Lai and Lee, 2006*). Embryos expressing the ΦC31 recombinase and harboring either the IMAGO recipient allele, VK00031 (UAS lines) or VK00018 (plot N^ICD) AttP landing site were microinjected in our lab or by companies (BestGene, GeneticServices). Selection for germline transformation was done by selecting either for the loss (IMAGO) or gain of *mini-white* activity (pUASTAttB). For the KI-lines, the orientation of the transgene insertion was determined by PCR. KI-lines with correct orientation were sequenced over the AttR sites to confirm proper integration of the transgenes.

## Fly stocks and husbandry

Fly stocks used in this study are: Canton-S, *w^1118*, *ato-KI* (this study), *amos-KI* (this study), *BfAth-KI* (this study), *MmAth1-KI* (this study), *MmAth5-KI* (this study), *PdAth2-KI* (this study), *AqbHLH1-KI* (this study), *tap-KI* (this study), *scute-KI* (this study), *UAS-ato* (this study), *UAS-amos* (this study), *UAS-cato* (this study), *UAS-BfAth* (this study), *UAS-BmAto* (this study), *UAS-MmAth1* (this study), *UAS-HsAth1* (this study), *UAS-MmAth5* (this study), *UAS-PdAth2* (this study), *UAS-AqbHLH1* (this study), *UAS-BmAto_opti* (this study), *UAS-PdAth2_opti* (this study), *UAS-atowbHLHMmAth1* (this study), *UAS-atowbHLHPdAth2* (this study), *UAS-MmAth1wbHLHato* (this study), *UAS-PdAth2wbHLHato* (this study), *UAS-scute* (this study), *plot-NICD* (this study), *dpp-GAl4, UAS-dGFP* (*Lieber et al., 2011*), *y^1 w^* P{10XUAS-IVS-mCD8::RFP}attP18 P{13XLexAop2-mCD8::GFP}su(Hw)attP8* (BDSC 32229) and *w^1118; P{GMR42A06-lexA}attP40* (BDSC 54268). Canton-S was used as our wild-type reference strain in all experiments except for the rhodopsin 6 staining where *w^1118* was used. All stocks were maintained at 25°C on standard cornmeal diet food, except where mentioned otherwise.

## Immunostainings

Fixation and immunostainings were performed following standard procedures. Briefly, adult retinas and imaginal discs were dissected in PBS, fixed with a 3.7% formaldehyde solution (in PBS) for 15 min (room temperature, RT), washed at least five times for 5 min with PBTr (PBS with 0.3% Triton X-100) (preceded by three washes in PBS for adult retinas). After blocking in PaxDG (1% BSA, 0.3% deoxycholic acid, 5% normal goat serum in PBTr) for 1 hr at RT, samples were incubated with primary antibodies in PaxDG overnight at 4°, washed again several times in PBTr, incubated in PaxDG with secondary antibodies for 1–2 hr at RT, washed again several times in PBTr and briefly washed in PBS before mounting. For anti-Ato stainings, PaxDG was replaced by a 3% BSA solution in PBTr. Embryos were dechorionated for 2 min in 50% bleach, washed in water, and fixed with a 1:1 proportion of heptane and 4% formaldehyde in PBS for 15 min at RT under heavy shaking. Collection of devitellinizionized embryos was performed by replacing the fixating solution by methanol or 90% ethanol (when phalloidin was used). After rehydration of the devitellinized embryos, immunostainings proceeded as described using PBTw (PBS 0,1% Tween-20) instead of PBTr and a 3% BSA, 5% goat serum solution in PBTw as blocking buffer. Embryos were briefly incubated in 50% glycerol before mounting.

Primary antibodies used in this study are listed in *Supplementary file 3*. Secondary antibodies conjugated with Alexa 488, Alexa 555 and Alexa 647 (Invitrogen) were used at 1:500. Alexa-coupled phalloidin (1:500; Invitrogen, A12379) was used together with the secondary antibodies. All samples were mounted in Vectashield (Molecular Probes, H-1200). Images were acquired at a Leica SP8 and

SP5 confocal microscopes and adjusted for brightness and contrast and maximum intensity projected in Fiji (*Schindelin et al., 2012*).

### *In situ* hybridization

Digoxigenin labeled RNA probes were synthesized in vitro using the DIG RNA Labeling Kit (SP6/T7) (Roche, 11175025910) following manual instructions. Probes were precipitated with LiCl and EtOH and stored in water at −20°.

The first steps were performed at RT. Imaginal discs were dissected in PBS, fixed in 3.7% formaldehyde in PBS for 20 min, fixed in 3.7% formaldehyde in PBTw for 20 min and washed several times in PBTw. Samples were incubated for 2 min in Proteinase K in PBTw (50 µg/µl), washed twice 1 min in glycine in PBTw (2 mg/ml), twice for 1 min each in PBTw, refixed for 20 min in 3.7% formaldehyde in PBTw and washed several times in PBTw. The following steps were performed at 55° with slight shaking. Samples were washed once for 10 min in 50% PBTw and 50% hybridization solution (HS, 50% formamide, 0.75 M NaCl, 0.075 M sodium citrate, 0.1% Tween 20, 100 µg/ml tRNAs, 50 µg/ml heparin in water), once in HS for 1 hr and incubated in HS with the RNA probe 1:500 overnight. Samples were washed twice in HS without heparin and tRNAs (HSw) for 20 min, for 20 min each in 75%, 50%, 25% HSw with PBTw. The remaining steps were performed at RT. Samples were washed again several times in PBTw, blocked with EBS, incubated with the alkaline phosphatase (AP) coupled anti-digoxigenin antibody for 1 hr in EBS, washed several times in PBTw, washed three times with AP buffer (100 mM NaCl, 50 mM MgCl$_2$, 100 mM Tris-HCl pH 9.5, 0.15 Tween 20 in water). In the dark, samples were incubated in a solution made out of NBT/BCIP tablets (Roche, 011697471001). Development of the staining was monitored periodically. Once the appropriate staining has been developed, samples were washed three times with PBTw and once with PBS. Discs were mounted in 80% glycerol. Images were acquired using a Leica DMRXA microscope controlled by Micro-Manager (*Edelstein et al., 2014*)

### ATH ectopic expression in wing imaginal discs

Flies were raised in density-controlled conditions (10 *UAS-ATH* virgin females and 10 *dpp-Gal4* males per vial, transferred every day). Vials with progenies that were laid on the 3rd, 4th, 5th and 6th day were allowed to develop for 10 days at 18°C, then kept at 25°C until eclosion (for the quantification of ectopic bristles in the adults) or raised at 25°C throughout development (for comparison of ATH expression in larval wing discs). Bristles along the midline until and including the anterior-cross vein of female wings were counted. Immunostainings were performed as described.

### qPCR

Third instar wandering larvae eye-antennal imaginal discs were dissected in RNAlater (Invitrogen, AM7020) and total RNA was extracted with RNAqueous Total RNA Isolation Kit (Ambion, AM1912) using standard procedures. cDNA was prepared with the QuantiTect Reverse Transcription Kit (Qiagen, 205310). The abundance of the different transcripts was measured with a LightCycler 480 SYBR Green I Master (Roche, 04707516001) on a LightCycler 480 instrument (Roche). The crossing points (CP) were calculated with the supplied program.

The linearity and the efficiency of amplification of each primer pair was determined by testing them on a dilution series of cDNA extractions. All primer pairs had a linearity of R-square >0.995 and the efficiency was included in calculation of the ratio using the following formula:

$$\frac{Target}{Reference} = \frac{\left(1 + Eff_{target}\right)^{-CP_{target}}}{\left(1 + Eff_{reference}\right)^{-CP_{reference}}}$$

The primer pairs listed in *Supplementary file 5* were used.

### Statistical analysis and plotting

The R-software package was used to plot the data and perform the indicated statistical tests (*R Core Team, 2015*).

## Stereomicroscopic images of adult eyes

Image stacks were acquired using a camera mounted on a stereomicroscope. Single focused images were reconstituted in FIJI using the 'Stack Focuser' plugin (https://imagej.nih.gov/ij/plugins/stack-focuser.html) (*Umorin, 2002*).

## Calculation of codon adaption index and synthesizing codon optimized genes

The codon adaptation index (CAI) was calculated online with the 'CAI Calculator 2' using 'Drosophila melanogaster-Carbone et al. 2003' as reference set (*Wu et al., 2005*). Codon optimization was done using the OPTIMIZER webtool with the *Drosophila melanogaster* entry in the Codon Usage Database as a reference for the codon optimization (*Nakamura et al., 2000*; *Puigbò et al., 2007*). Sequences (*Supplementary file 7*) were manually modified to fit the production constraints. Genes were synthesized at Integrated DNA Technologies.

## Scanning electron microscopy and counting ommatidia

20 flies were grown in density-controlled conditions. Freshly hatched females were washed in PBS and fixed in 50% PBS with 4% formaldehyde and 50% ethanol (EtOH) for several hours at room RT. Samples were dehydrated by washing them twice for 10 min in each of 70%, 90% EtOH with water and 100% EtOH. All solutions were sterile filtered prior to use. Upon this, samples were washed twice in HMDS for 30 min at RT and placed into a desiccator overnight for drying. Samples were mounted on carbon stickers with silver paint and coated with chrome in a Leica EM ACE600. Images were acquired at a ZEISS SIGMA Variable Pressure Field Emission-Scanning Electron Microscope. Images were adjusted for brightness and contrast in Fiji (*Schindelin et al., 2012*).

## Analysis of the JO

Flies were raised on standard medium in incubators maintained at 25°C and 60% relative humidity, with a 12 hr:12 hr light:dark cycle. Male mutants were identified on the day of eclosion using $CO_2$ sedation and allowed to age in separate vials – experiments were conducted on 2–6 day-old flies at temperatures 21–23°C. Flies were mounted as described previously (*Albert et al., 2007*). Briefly, male flies were attached, ventrum-down, to the head of a Teflon rod using blue light-cured dental glue. The second antennal segment of the antenna under investigation was glued down to prevent movement. A vibration isolation table was used. After mounting, flies were placed inside a rectangular steel chamber (6 × 6 × 2.5 cm) attached to a micromanipulator on top of the vibration isolation table. The chamber was situated perpendicular to the Laser Doppler Vibrometer (LDV) and had a porthole on the side facing the LDV so that the laser beam could still be focused on the tip of the arista. The fly was placed at the bottom of the chamber in a central position, with the antenna perpendicular to the laser beam. A small plastic case (3.5 × 2.5 × 2.5 cm) was placed over the fly. $CO_2$ was introduced to a cavity underneath the chamber via a plastic tube. Flow rate was kept constant using a flow regulator (Flowbuddy, Flystuff). Before $CO_2$ was introduced to the chamber a free fluctuation recording was taken to assess the baseline level of antennal function. The chamber was then flooded with $CO_2$ (BOC, 99.5% purity) for 1 min at a constant flow rate of 3 l/min. Immediately after $CO_2$ flow was extinguished and another free fluctuation was taken to record the antennal fluctuations in the passive state. The fly was then allowed 5 min to recover from the sedation before a final free fluctuation recording was taken. Only flies that were able to recover a sufficient level of antennal function were analyzed.

Electrostatic stimulation was evoked using two external actuators positioned close to the antenna's arista (for details see [*Effertz et al., 2012*]). Two electrodes were inserted into the fly – a charging electrode was placed into the thorax so that the animal's electrostatic potential could be raised to 20 V against ground, and a recording electrode for measuring mechanically evoked compound action potentials (CAPs) was introduced close to the base of the antenna which was not glued down. The charging electrode was also used as reference electrode for the CAP recordings.

Arista displacements were measured at the arista's tip using a PSV-400 LDV with an OFV-70 close up unit (70 mm focal length) and a DD-5000 displacement decoder. The displacement output was sampled at a rate of 100 kHz using a CED 1401 A/D converter and the Spike 2 software (both Cambridge Electronic Design Ltd., Cambridge, England). Free (i.e. unstimulated) fluctuations of the

arista were measured both before and after the experiment to judge potential changes to the antennal system. Only those flies which maintained reasonable levels of antennal function throughout the experiment were analyzed.

## Acknowledgements

We would like to thank A Salzberg, C Desplan, A Jarman, H Bellen, Y Hiromi, S Crews and the Developmental Studies Hybridoma Bank (and their contributors listed in *Supplementary file 3*) for antibodies, F Pignoni, L Holland, N Brown, M Vervoort, B Degnan for gDNA or vectors, G Struhl for the UAS dGFP stock, the Electron Microscopy Platform of the Center for Human Genetics (KU Leuven-VIB) and P. Baatsen for help with SEM and the PICPS imaging facility of the ICM. We also would like to thank N Mora and R Ejsmont for transgenic flies, all members of the Hassan lab for critical discussion and support and G Halder and S Aerts for critical input. The Leica SP8x confocal microscope was provided by InfraMouse (KU Leuven-VIB) through a Hercules type 3 project (ZW09-03). The research leading to these results has received funding from the program "Investissements d'avenir" ANR-10- IAIHU-06, VIB, the WiBrain Interuniversity Attraction Pole (BELSPO IUAP) network, Fonds Wetenschappelijke Onderzoeks (FWO) grants G.0543.08, G.0680.10, G.0681.10 and G.0503.12 (BAH). SW was member of the Marie Curie initial training network 'FLiACT' funded under the seventh Framework Program by the European Union. MPT received funding from the Engineering and Physical Sciences Research Council (EP/F500351/1) through UCL CoMPLEX and JTA was supported by grants from the Human Frontier Science Program (RGY0070/2011) and the Biotechnology and Biological Sciences Research Council (BB/L02084X/1). BAH is an Allen Distinguished Investigator and an Einstein Fellow of the Berlin Institute of Health.

## Additional information

### Funding

| Funder | Author |
| --- | --- |
| Vlaams Instituut voor Biotechnologie | Simon Weinberger<br>Jiekun Yan<br>Annelies Claeys<br>Natalie De Geest<br>Duru Ozbay<br>Bassem A Hassan<br>Ariane Ramaekers |
| Fonds Wetenschappelijk Onderzoek | Simon Weinberger<br>Jiekun Yan<br>Annelies Claeys<br>Natalie De Geest<br>Duru Ozbay<br>Bassem A Hassan<br>Ariane Ramaekers |
| Federaal Wetenschapsbeleid | Simon Weinberger<br>Jiekun Yan<br>Annelies Claeys<br>Natalie De Geest<br>Duru Ozbay<br>Bassem A Hassan<br>Ariane Ramaekers |
| European Commission | Simon Weinberger<br>Bassem A Hassan |
| Human Frontier Science Program | Matthew P Topping<br>Talah Hassan<br>Xiaoli He<br>Joerg T Albert |
| Biotechnology and Biological Sciences Research Council | Matthew P Topping<br>Talah Hassan<br>Xiaoli He<br>Joerg T Albert |

| Paul G. Allen Family Foundation | Bassem A Hassan<br>Ariane Ramaekers |
|---|---|
| Einstein Stiftung Berlin | Bassem A Hassan |
| Institut Hospitalo-Universitaire<br>IHU-A-ICM | Ariane Ramaekers<br>Bassem A Hassan |
| Institut du Cerveau et de la<br>Moelle épinière | Ariane Ramaekers<br>Bassem A Hassan |

The funders had no role in study design, data collection and interpretation, or the decision to submit the work for publication.

### Author contributions
SW, MPT, Conceptualization, Investigation, Methodology, Writing—original draft, Writing—review and editing; JY, AC, NDG, DO, TH, XH, Methodology, Acquisition of data; JTA, BAH, Conceptualization, Supervision, Funding acquisition, Writing—original draft, Writing—review and editing; AR, Conceptualization, Supervision, Investigation, Methodology, Writing—original draft, Writing—review and editing

### Author ORCIDs
Bassem A Hassan, http://orcid.org/0000-0001-9533-4908
Ariane Ramaekers, http://orcid.org/0000-0002-3548-774X

## Additional files

### Supplementary files
• Supplementary file 1. Organisms for tBLASTn search.

• Supplementary file 2. Sources of DNA.

• Supplementary file 3. Antibodies used in this study.

• Supplementary file 4. Codon adaptation index (CAI) of the genes used in this study.

• Supplementary file 5. Primer pairs used for qPCR.

• Supplementary file 6. (A) Summary statistics of *Figure 2D*; number of observations (N), mean, median, standard deviation, 95% confidence interval and p-value of Shapiro Wilk test. (B) p-values of *Figure 2D*; p-values (t-test) and adjusted p-values (by Holm method). (C) Summary statistics of *Figure 3B*; number of observations (N), mean, median, standard deviation, 95% confidence interval and p-value of Shapiro Wilk test. (D) p-values of *Figure 3B*; p-values (Fisher exact test) and adjusted p-values (by Holm method). (E) Summary statistics of *Figure 5B* for bristles; number of observations (N), mean, median, standard deviation, 95% confidence interval and p-value of Shapiro Wilk test. (F) Summary statistics of *Figure 5B* for campaniform organs; number of observations (N), mean, median, standard deviation, 95% confidence interval and p-value of Shapiro Wilk test. (G) p-values of *Figure 5B* for bristles; p-values (Wilcoxon Rank Sum and Signed Rank Tests) and adjusted p-values (by Holm method). (H) p-values of *Figure 5B* for campaniform organs; p-values (Wilcoxon Rank Sum and Signed Rank Tests) and adjusted p-values (by Holm method) (I) Summary statistics of *Figure 5C* for bristles; number of observations (N), mean, median, standard deviation, 95% confidence interval and p-value of Shapiro Wilk test. (J) Summary statistics of *Figure 5C* for campaniform organs; number of observations (N), mean, median, standard deviation, 95% confidence interval and p-value of Shapiro Wilk test. (K) p-values of *Figure 5C* for bristles; p-values (Wilcoxon Rank Sum and Signed Rank Tests) and adjusted p-values (by Holm method). (L) p-values of *Figure 5C* for campaniform organs; p-values (Wilcoxon Rank Sum and Signed Rank Tests) and adjusted p-values (by Holm method). (M) Summary statistics of *Figure 5D* for bristles; number of observations (N), mean, median, standard deviation, 95% confidence interval and p-value of Shapiro Wilk test. (N) Summary

statistics of *Figure 5D* for campaniform organs; number of observations (N), mean, median, standard deviation, 95% confidence interval and p-value of Shapiro Wilk test. (O) p-values of *Figure 5D* for bristles; p-values (Wilcoxon Rank Sum and Signed Rank Tests) and adjusted p-values (by Holm method). (P) p-values of *Figure 5D* for campaniform organs; p-values (Wilcoxon Rank Sum and Signed Rank Tests) and adjusted p-values (by Holm method). (Q) Summary statistics for passive antennal mechanics (best frequency, f0; tuning sharpness, Q; apparent mass) across ATH rescues (relating to *Figure 4A*); number of observations (N), mean, median, standard deviation, standard error, 95% confidence interval and p-values for pair-wise comparison with Ato KI control (t-test for normally distributed data and Mann-Whitney Rank Sum test (MWRS) for non-normally distributed data. Significances are highlighted in color (<0.05 and<0.001). (R) Summary statistics for active antennal mechanics (best frequency, f0; tuning sharpness, Q; energy gain) across ATH rescues (relating to *Figure 4A*); number of observations (N), mean, median, standard deviation, standard error, 95% confidence interval and p-values for pair-wise comparison with Ato KI control (t-test for normally distributed data and Mann-Whitney Rank Sum test (MWRS) for non-normally distributed data. Significances are highlighted in color (<0.05 and<0.001). (S) Summary statistics for gating compliance analysis (relating to *Figure 4—figure supplement 1B*) across ATH rescues (number of sensitive ion channels, Ns; number of insensitive ion channels, Ni; single channel gating force of sensitive ion channels, zs; single channel gating force of insensitive ion channels, zi; asymptotic stiffness, Kinf; steady state stiffness, Ksteady; total gating spring stiffness, KGS); number of observations (N), mean, median, standard deviation, standard error, 95% confidence interval and p-values for pair-wise comparison with Ato KI control (t-test for normally distributed data and Mann-Whitney Rank Sum test (MWRS) for non-normally distributed data. Significances are highlighted in color (<0.05 and<0.001). Note that parameter values for the PdAth2-KI were dispensed from statistical comparison to the control condition, as the transduction system in the antennae of PdAth2-KI flies did not comply with the two transducer population model (from *Effertz et al., 2012*), but rather conformed to a one transducer population model. It was thus not immediately evident how to compare the single transducer populations of PdAth2-KI flies to the two transducer (sensitive and insensitive, respectively) populations of control flies.

• Supplementary file 7. Sequences codon optimized genes.

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
