## [Decision Letter]

Thank you for choosing to send your work, "Evolutionary changes in proneural coding sequence quantitatively regulate sensory organ development and function", for consideration at *eLife*. Your article has been favorably evaluated by K VijayRaghavan (Senior Editor) and three reviewers, one of whom is a member of our Board of Reviewing Editors. Although the work is of interest, we regret to inform you that the findings at this stage are too preliminary for further consideration at *eLife*.

All three reviewers comment positively on the quality of the data. The descriptive data are very nice but the three reviewers also commented on the lack of insight at the mechanistic level. There reviews are summarized below.

Summary

In this manuscript, Weinberger and co-authors report a study about the functional evolution of Atonal/ATH family genes, which encode basic Helix-Loop-Helix (bHLH) transcription factors with key roles during neurogenesis in *Drosophila* and vertebrates. The authors mostly studied the effects in *Drosophila* of the expression of ATH genes from several other species. Such interspecific expression experiments are not new, including for bHLH genes, and some of the previous studies have proven to be much insightful. The present study nevertheless uses much improved methodologies as compared to previous studies. First, in contrast to earlier studies that were based on ectopic expressions driven by UAS/Gal4 or heat-shock promoter, in this study, the authors mainly used knock-ins in which the coding region of *Drosophila*atonal was replaced by the coding region of other genes. Second, whereas previous published works were usually centered on the study of a single gene (or a few genes) from a single species, here several genes of the families from several distantly-related species were studied. Finally, the effects of the different ATH genes were assessed on several different sense organs, rigorous quantitative analyses were performed and in one case (Johnston's organ) the function of the organs was studies. The authors found that the ATH genes, but not the non-ATH genes (with the exception of a sponge gene that is equally related to ATH and some non-ATH genes such as neurogenin and NeuroD) were able to functionally replace atonal. A generally consistent pattern emerges, with Atonal and amos giving the best results, followed by mammalian Atoh1 and then other atonal homologues, with the *Platynereis* Ath2 homologue generally having the lowest function. In addition, the authors test the "proneural potency" of different atonal homologues in an over-expression assay in the wing imaginal disc. Here, mouse Atoh1 is most potent, followed by atonal itself, with *Platynereis* Ath2 again having one of the lowest activities. The authors correlate this potency with the protein stability/perdurance of several atonal homologues (atonal, amos, cato and mouse Atoh1) for which antibodies are available. Nevertheless, quantitative analysis showed that the different genes have different activities and most of the knock-in genes behaves like weak hypomorphic mutations of atonal. The authors also showed that the different genes have different “proneural potencies” that may correlate with the temporal perdurance of ATH proteins.

This is an original, interesting, well-designed and well-conducted study that provides new and important insights in the evolution of neurogenic bHLH genes. Overall the manuscript is well written and very clear. We are impressed by the overall high quality of the data, in particular those about the knock-ins and I agree with most of the interpretations made by the authors.

The model proposed by the authors in which functional differences would be due to different protein dynamics, is interesting, but is based on correlations involving only very few proteins (those for which antibodies do exist) and is therefore not well substantiated. I think that the authors should try to provide some further evidence in favor of their model.

My main concern with the paper is that Figure 1–Figure 5 are entirely descriptive, and the only mechanistic explanation for the data is based on the intensity of protein staining of a small subset of the atonal homologues in Figure 6, and this only applies to the gain-of-function assays, not the gene replacement experiments. Thus, while the clear differences in function between the atonal homologues is established beyond doubt, the mechanism underlying these differences is still vague. One possible way of addressing this would be to repeat the gene replacement experiments with homologues that are tagged with GFP to directly measure protein levels and stability for all homologues. This form of C-terminal tagging has worked well for mouse Atoh1, but the time required for such experiments place it beyond the scope for rebuttal in an *eLife* submission.

The previous comment is reinforced by a comment of other reviewers. Why do the authors put so much focus on auto regulation as they did not perform experiments that address this question?

The bHLH domain swaps experiments are really puzzling and these experiments raise many questions. Yet, they are not discussed and not explored. Are the chimeric genes less transcribed, or give rise to less stable mRNAs or proteins? Did the authors compare mRNA and protein (when possible with the available antibodies) levels produced by the chimeric and “normal” genes?

---

## [Author Response]

*[…] The model proposed by the authors in which functional differences would be due to different protein dynamics, is interesting, but is based on correlations involving only very few proteins (those for which antibodies do exist) and is therefore not well substantiated. I think that the authors should try to provide some further evidence in favor of their model.*

*My main concern with the paper is that Figure 1–Figure 5 are entirely descriptive, and the only mechanistic explanation for the data is based on the intensity of protein staining of a small subset of the atonal homologues in Figure 6, and this only applies to the gain-of-function assays, not the gene replacement experiments. Thus, while the clear differences in function between the atonal homologues is established beyond doubt, the mechanism underlying these differences is still vague. One possible way of addressing this would be to repeat the gene replacement experiments with homologues that are tagged with GFP to directly measure protein levels and stability for all homologues. This form of C-terminal tagging has worked well for mouse Atoh1, but the time required for such experiments place it beyond the scope for rebuttal in an eLife submission.*

We sought to strengthen mechanistic insight by addressing the reviewer’s suggestion of generating tagged-ATHs. We also performed an additional experiment to identify the genetic mechanism behind the differential stability of ATHs. Together these experiments provide additional support for our model of differential protein stability and identify the differential post-transcriptional sensitivity to Notch-mediated lateral inhibition as the likely operant mechanism. Below we detail our new experiments and their results.

A) We generated C-terminal GFP tagged constructs and corresponding KI-lines for Ato, MmAth1, MmAth5 and PdAth2t (acting as “strong” or “weak” proneural factors in the fruit fly). We then examined GFP expression in eye-antennal discs and adult eye morphology. The results are described in the revised manuscript (Figure 6—figure supplement 2). To summarize, we find that the C-terminal tags differentially affect the functional properties of the ATHs. For instance, while Ato::GFP behaves similarly to non-tagged Ato KI and appears normal in terms of expression and phenotype, even a single copy of MmAth1::GFP causes lethality at pupal stage, preventing the establishment of the stock. Surprisingly, MmAth5::GFP behaves similarly to non-tagged MmAth1, both in terms of expression in the eye-antennal disc and adult eye morphology. MmAth5::GFP expression outlives that of Ato::mCherry, extending posteriorly and showing reduced R8 singling out. The adult eye of MmAth5::GFP shows a severe rough phenotype and necrosis spots, similarly to untagged MmAth1 (Figure 6—figure supplement 2). We draw two main conclusions from these data. First, not all ATHs can be tagged – at least C-terminally with GFP – suggesting the need for caution in interpreting data from tagged transgenes in general. Second, the correlation between the MmAth1-like phenotype of the MmAth5::GFP KI-line and its MmAth1-like protein expression pattern provides additional support to the notion that proneural potency changes between ATHs derive from differential protein steady state dynamics.

B) Recent data from several models suggests that Notch signaling can regulate proneural protein function post-transcriptionally at the level of protein stability (Sriuranpong et al., 2002; Qu et al., 2013; Kiparaki et al., 2015). Interestingly, the patterns of ATH proteins upon ectopic expression in the wing (Figure 6) were reminiscent of what is expected upon lateral inhibition, namely single cells expressing an ATH surrounded by ATH-negative cells. This was the case for Ato, for example, but not for MmAth1. Since ATH expression in these cases is driven by a heterologous promoter (the Gal4/UAS system) this differential regulation cannot be transcriptional. This is similar to what we observed with the KI-lines in the eye disc where the mRNAs of Ato and MmAth1 are similar, but the MmAth1 protein lasts longer (Figure 6; also, see below). We thus tested whether ATH protein expression levels observed upon ectopic expression could be modulated by Notch, independently of any effect on transcript levels (Figure 6). Upon activation of the Notch pathway, Amos, Ato and Scute but not MmAth1 protein levels dropped suggesting that (i) ATH protein levels can be regulated by Notch signaling and (ii) that distinct ATHs present different sensitivities to this Notch-dependent regulation. This is consistent with the observation that MmAth1 protein expression in MmAth1 KI eye-antennal discs is characterized by a homogeneous pattern, devoid of precursor cell singling out.

We believe that, together, these experiments strongly support our model and identify proneural potency as a differential response to Notch signaling activity.

“My main concern with the paper is that Figure 1–Figure 5 are entirely descriptive, and the only mechanistic explanation for the data is based on the intensity of protein staining of a small subset of the atonal homologues in Figure 6, and this only applies to the gain-of-function assays, not the gene replacement experiments.”

While it is true that in a first set of experiments in Figure 6 is indeed based on anti-ATH immunostainings on wing discs upon ectopic expression, the rest of the figure shows comparable results based on endogenous KIs. Importantly, the KI and the ectopic expression data agree with each other. Ectopic expression experiments revealed that, despite an identical transcriptional regulation imposed by the GAL4/UAS system, the number of ATH – positive nuclei as compared to a co-expressed destabilized GFP (dGFP) reporter varies between ATHs (Figure 6). However, we also performed *in situ* hybridizations and immunostainings on KI-lines (Figure 6) showing that while Ato and MmAth1 mRNA present a comparable expression pattern, MmAth1 protein expression posterior to the MF was more homogeneous and stronger as compared to Ato. Finally, we attempted to create tagged KIs for multiple ATHs and found that the tags interfere with the function of some of them (see response to comment 1 above for details). Interestingly, at least in one case the GFP tag seemed to stabilize the ATH, and result in a “gain of function”-like phenotype.

*Why do the authors put so much focus on auto regulation as they did not perform experiments that address this question?*

Autoregulation is essential for ensuring proper expression and function of endogenous atonal (Baker et al., 1996; Sun et al., 1998). Thus, variation in autoregulation – potentially caused by different affinities for endogenous*ato* regulatory sequences – was an obvious candidate to explain the functional differences between Ato homologues. We do in fact address this issue experimentally in two different ways in the original manuscript. First, we performed qRT-PCR experiments that revealed high *ATH* mRNAs expression levels in the eye-antennal discs (Figure 5—figure supplement 1), demonstrating that autoregulation does take place in every KI-line. These experiments also showed no correlation between *ATH* mRNA levels and the corresponding proneural potency, indicating that autoregulation alone is unlikely to explain the functional differences between ATHs. Second, by driving their ectopic expression using the GAL4/UAS system, we investigated functional differences between ATHs independently of possible changes in transcriptional regulation, including autoregulation. In summary, autoregulation is addressed experimentally and is ruled out as a possible mechanistic explanation for differential proneural potency.

*The bHLH domain swaps experiments are really puzzling and these experiments raise many questions. Yet, they are not discussed and not explored. Are the chimeric genes less transcribed, or give rise to less stable mRNAs or proteins? Did the authors compare mRNA and protein (when possible with the available antibodies) levels produced by the chimeric and “normal” genes?*

We agree with the reviewer that the swaps experiments are interesting but we feel that further detailed investigation is not within the scope of this study. This is for two main reasons. First, the swaps were done to specifically explore whether the bHLH domain alone “encodes” proneural potency, and we think we can safely conclude that this is not the case. Second, investigating the precise reason why these – ultimately artificial – chimeras are generally less active than their cognate proteins, although some show intermediate activity, may be interesting, but does not add much to understanding how the natural proteins are regulated at this stage, and thus provide little insight into the fundamental problem addressed in this work, namely the potential contribution of ATH CDS evolution.